# Virulence Reduction in *Yersinia pestis* by Combining Delayed Attenuation with Plasmid Curing

**DOI:** 10.3390/biom16010040

**Published:** 2025-12-25

**Authors:** Svetlana V. Dentovskaya, Rima Z. Shaikhutdinova, Mikhail E. Platonov, Nadezhda A. Lipatnikova, Elizaveta M. Mazurina, Tat’yana V. Gapel’chenkova, Pavel Kh. Kopylov, Sergei A. Ivanov, Alexandra S. Trunyakova, Anastasia S. Vagaiskaya, Andrey P. Anisimov

**Affiliations:** Laboratory for Plague Microbiology, Especially Dangerous Infections Department, State Research Center for Applied Microbiology and Biotechnology, 142279 Obolensk, Russia; dentovskaya@obolensk.org (S.V.D.); shaikhutdinova@yandex.ru (R.Z.S.); platonovmichael@mail.ru (M.E.P.); n.a.lipatnikova@mail.ru (N.A.L.); elizavetamazurina99@yandex.ru (E.M.M.); tgapelchenkova@mail.ru (T.V.G.); pch.kopylov@gmail.com (P.K.K.); sa-ivanov@yandex.ru (S.A.I.); sasha_trunyakova@mail.ru (A.S.T.); vagaiskaya.anastasiya@gmail.com (A.S.V.)

**Keywords:** *Yersinia pestis*, Crp, regulated delayed attenuation, *crp*, plague vaccine, protection

## Abstract

*Yersinia pestis* caused the three plague pandemics that claimed more than two hundred million human lives. There is still no vaccine that meets all WHO requirements, and many researchers continue to develop plague vaccines using various technological platforms. For example, researchers led by Roy Curtiss 3rd have developed a new approach to achieve controlled, delayed attenuation of bacterial pathogens. Mutants generated using this method were superior in protecting *Y. pestis*-infected mice immunized with strains generated using traditional gene knockout. However, further studies are needed to determine the safety and efficacy of these delayed-attenuated strains in other mammalian species in order to extrapolate on humans the data obtained in accordance with the FDA Animal Rule. Three *Y. pestis* strains, a Δ*crp* mutant, a mutant with arabinose-dependent regulated *crp* expression (*araC* P_BAD_ *crp*) or an *araC* P_BAD_ *crp* mutant cured of plasmid pPst were derived from virulent wild-type strain 231. To evaluate the safety, outbred mice or guinea pigs were immunized subcutaneously with serial tenfold dilutions of mutated strains. For vaccine studies, immunized animals were subcutaneously challenged with 200 LD_100_ (lethal dose in all exposed subjects) of the wild-type *Y. pestis* strain. The challenge caused the death of 100% of naïve animals in controls. The *Y. pestis* strain 231Δ*crp* was nonlethal in mice at a dose of 10^7^ CFs. The LD_50_ of the 231Δ*crp* strain in guinea pigs increased by at least 10^7^-fold compared to that of the wild-type strain. The LD_50_s of the 231P_BAD_-*crp* mutant in mice and guinea pigs were approximately 10^4^-fold and 10^7^-fold higher than those of *Y. pestis* 231, respectively. The 231P_BAD_-*crp*(pPst¯) strain did not cause death in mice (LD_50_ > 10^7^ CFU) and guinea pigs (LD_50_ > 10^9^ CFU) when administered subcutaneously and was capable of inducing intense protective immunity in both species of laboratory animals. Our research has shown once again the necessity of balance between safety and effectiveness demonstrating the feasibility of further investigation of *crp* mutants as promising candidate plague vaccines.

## 1. Introduction

Plague is a disease of wild rodents, which only accidentally crosses over to humans [1]. However, these accidents were the cause of at least three known pandemics and cost humanity more than 200 million lives [2,3]. The World Health Organization (WHO), a key player in pandemic preparedness and response [4], has released a list of prioritized pathogens posing the greatest emerging epidemic threats. WHO draws the attention of the global community to the need to accelerate research in the development and clinical trials of vaccines against these priority pathogens [5,6]. This list contains five bacterial pathogens, including *Yersinia pestis*, the causative agent of the plague. Although the success in the fight against the third plague pandemic was largely due to the application of the first generation of plague vaccines (e.g., a killed plague vaccine developed in 1897 by Waldemar Mordecai Haffkine (1860–1930)) [7,8] there is still no generally accepted plague vaccine that meets all WHO requirements [9]. A number of researchers continue to develop plague vaccines using various technological platforms [10,11]. Advances in molecular bacteriology have made it possible to develop a wide range of plague vaccine candidates that appear to be superior to the first-generation vaccines in vitro and in animal comparative studies [12]. Recently, Roy Curtiss 3rd’s team proposed an original methodology for controlled delayed attenuation of bacterial pathogens [13,14,15]. This technology has been applied to *Y. pestis* strain KIM5+ (subsp. *pestis* bv. Medievalis) in a mouse model and has shown superiority over traditional knockout mutagenesis [16]. However, the safety and efficacy of the strains with delayed attenuation in humans is not yet known.

Currently, ethical considerations prohibit the use of standard research methodologies like randomized controlled trials or cohort studies for evaluating plague vaccines. This is because failure to vaccinate the control group would expose its members to unacceptable risk. Therefore, the authorization of plague vaccines for use in humans should be conducted utilizing the FDA’s Animal Rule [17]. This rule facilitates the licensing of products for which traditional potency testing methodologies are deemed unethical. Specifically, the Animal Rule is applied in situations where an infective agent exhibits extremely high pathogenicity and the rarity of the infection it causes precludes the feasibility of conducting field studies (21 CFR 601.91 Subpart H, “Approval of Biological Products When Human Efficacy Studies Are Not Ethical or Feasible”) [18]. One of the four essential requirements for the application of the FDA Animal Rule is that the vaccine’s effects (survival or prevention of serious disease) must be observed in more than one animal species, which is expected to be more likely to reflect human responses.

The aim of this study was to verify the vaccine properties of the P_BAD_-*crp* mutant generated on the basis of a phylogenetically different parent strain and to evaluate its properties not only in mice but also in guinea pigs.

## 2. Materials and Methods

### 2.1. Bacterial Strains, Plasmids, and Culture Conditions

All bacterial strains and plasmids used in this study are listed in Table 1. *Escherichia coli* DH5α was used for plasmid propagation. *Y. pestis* cells were grown routinely at 28 °C in BHI, HiMedia, Mumbai, India) or on BHI agar at pH 7.2. *E. coli* cells were grown routinely at 37 °C in Luria–Bertani (LB) broth (HiMedia, Mumbai, India) or on agar plates at pH 7.2. Ampicillin (100 µg/mL, Amp) (neoFroxx, Einhausen, Germany), chloramphenicol (20 µg/mL, Cm) (neoFroxx, Einhausen, Germany), polymyxin B (50 μg/mL, Pol) (neoFroxx, Einhausen, Germany), L-arabinose (200 µg/mL, Ara) (HiMedia, Dahej, India) were supplemented as appropriate. BHI agar containing 10% sucrose was used for sacB-based counterselection in the allelic exchange experiments. MacConkey agar (HiMedia, Mumbai, India) with 0.5% maltose (HiMedia, Dahej, India) was used to indicate sugar fermentation and *crp* expression.

### 2.2. Animals

Six-week-old outbred mice (Lab of Biomodels breeding unit, SCRAMB, Moscow, Russia) and four-week-old guinea pigs («Andreevka» branch of the Scientific Center of Biomedical Technologies of the Federal Medical and Biological Agency (Andreevka, Moscow, Russia) both male and female were used in all studies. All animal experiments were approved by the State Research Center for Applied Microbiology and Biotechnology (Obolensk, Moscow, Russia) Bioethics Committee (Permit No: VP-2025/1 on 30 April 2025) and were performed in compliance with the NIH Animal Welfare Insurance #A5476-01 issued on 2 July 2007, and the European Union guidelines and regulations on handling, care and protection of Laboratory Animals (https://environment.ec.europa.eu/topics/chemicals/animals-science_en (accessed on 18 November 2025).

### 2.3. Construction of the Plasmids

The primers used in this work are listed in Table 2.

A primer set of Crp-NdeI/Crp-XhoI was used for amplifying the gene sequence of *crp*. The resulting PCR product was digested with NdeI and XhoI and ligated with plasmid pET-24b(+) digested with the same restriction enzymes to construct plasmid pET24-crp. *E. coli* BL21(DE3) was transformed with the plasmid pET24-crp.

Primer sets of Crp-HindIII/Crp-SphI and Crp-XbaI/Crp-SalI were used for amplifying URcrp (upstream region gene sequence of *crp*, from −664 to −110 bp) and the gene sequence of *crp* with the Shine-Dalgarno sequence, from −36 to +633 bp, respectively. A primer set of Pbad-SphI/Pbad-XbaI was used for amplifying the *araC* P_BAD_ promoter from strain *E. coli* DH5α. The resulting PCR products were digested with appropriate restriction enzymes and ligated with pUC57 digested with HindIII and SalI to construct the plasmid pUC57-URcrp-araC Pbad-crp.

A primer set of Lt0-SphIF/Lt0-SphIR was used for amplifying the transcription terminator t0 from bacteriophage λ (Lt0TT). The resulting PCR product was digested with SphI and ligated with the plasmid pUC57-URcrp-araC Pbad-crp digested with the same restriction enzyme to construct the plasmid pUC57-URcrp-Lt0TT-araC Pbad-crp. The plasmid pUC57-URcrp-Lt0TT-araC Pbad-crp was digested with SmaI and ligated with the suicide vector pCVD442 digested with the same restriction enzyme to construct the plasmid pCVD442-URcrp-Lt0TT-araC Pbad-crp.

All the plasmid constructions were verified through sequencing.

### 2.4. Mutagenesis

The *crp* deletion was constructed in the *Y. pestis* EV strain by λRed-mediated mutagenesis [21] and confirmed by PCR (Table 2). The *Y. pestis* EVΔ*crp*::*cat* DNA fragment containing the respective deletion was then subcloned into the pCVD442 plasmid [22]. The pCVD442-Δ*crp*::*cat* plasmid was transferred from an *E. coli* S17-1 *λpir* donor strain into a recipient wild type *Y. pestis* strain 231 by conjugation. Elimination of the suicide vector and selection of *Y. pestis* clones were performed in the presence of 5% sucrose and chloramphenicol [22]. The resistance cassette was eliminated to produce FRT scar mutants by introducing pCP20 [21]. 

The procedure for constructing the *Y. pestis* 231P_BAD_-*crp* mutant was performed using the standard method [22]. In brief, the suicide plasmid pCVD442-URcrp-Lt0TT- *araC* P_BAD_-*crp* was conjugationally transferred from *E. coli* S17-1 λpir to *Y. pestis* 231. A single-crossover insertion strain was isolated on BHI agar plates containing ampicillin. Loss of the suicide vector after the second recombination between the homologous regions was selected using the *sacB*-based sucrose sensitivity counter-selection system. The colonies were screened for Amp^S^ and verified by PCR using the primers Crp-KF/Crp-KR.

The isogenic pPst-plasmid-cured strain of *Y. pestis* 231P_BAD_-*crp*(pPst¯) was generated as described earlier. It was cloned from the 231P_BAD_-*crp* strain after 10 serial passages caused in BHI broth media at +5 °C [24]. Clones lacking the 9.5-kb pPst plasmid were detected by PCR and confirmed by a fibrinolysis assay [25].

The presence of all *Y. pestis* resident plasmids [26,27,28,29] was confirmed via PCR amplification.

### 2.5. Preparation of Crp Antiserum

Full-length His-tagged Crp was expressed from *E. coli* BL21(DE3)/pET24-*crp.* Briefly, recombinant *E. coli* culture was grown to log phase in LB broth at 37 °C with agitation. Cells were then induced with 1 mM IPTG (isopropyl-β-d-thiogalactopyranoside) for 4 h. After sonication and centrifugation, protein was solubilized in the binding buffer containing 6 M urea and was purified in an affinity chromatography using Ni^2+^-Sepharose chromatography under denaturing conditions.

Mouse anti-CRP serum was collected after two subcutaneous immunizations with 20 μg per animal after being mixed with the Imject Alum (1:1).

### 2.6. Western Blotting

The whole-cell lysates were separated by sodium dodecyl sulfate-polyacrylamide (10%) gel electrophoresis and then electrophoretically transferred to a nitrocellulose membrane using equipment and protocols from Bio-Rad Laboratories (Hercules, CA, USA). The membrane was blocked overnight at 4 °C with 5% (*w*/*v*) nonfat dry milk in phosphate-buffered saline containing 0.2% Tween 20 (PBS-T), washed three times for 10 min with PBS-T, probed at room temperature for 1.5 h with a mice polyclonal against His-tagged Crp from *E. coli* BL21(DE3)/pET24-*crp* diluted in PBS-T, and then washed three times with PBS. Immunoreactive proteins were detected using a horseradish peroxidase-conjugated anti-mouse IgG-HRP (1:5000, Sigma, Saint-Louis, MO, USA). Proteins were visualized with DAB substrate kit (Abcam, Cambridge, UK) and photographed using an Amersham ImageQuant 800 (Cytiva, Wilmington, NC, USA) gel imaging system.

### 2.7. Animal Challenges

*Y. pestis* strains were grown at 28 °C for 48 h on BHI plus L-arabinose (200 µg/mL), diluted to an appropriate concentration in PBS. The challenge doses were determined by serial dilutions in PBS and plating on BHI agar.

To demonstrate the loss of virulence, mice (*n* = 6/per dose) and guinea pigs (*n* = 6/per group) were challenged subcutaneously (s.c.) with the 231Δ*crp*, 231P_BAD_-*crp* or 231P_BAD_-*crp*(pPst¯) strains. Each strain was inoculated at different 10-fold doses at 0.2 mL aliquots (10^2^ CFU–10^7^ CFU per mouse, 10^2^–10^9^ CFU per guinea pig). At the same time, 4 additional groups of 6 mice or guinea pigs were used to control the virulence of the wild type *Y. pestis* 231. Then the animals were monitored for 28 days to assess the survival rate.

For vaccine studies, groups of 6 outbred mice or 6 guinea pigs were vaccinated via the subcutaneous (s.c. route with serial dilutions of the *Y. pestis* 231 strain P_BAD_-*crp*(pPst¯) or only with PBS buffer as a placebo. Blood was collected on day 28 postimmunization for antibody measurement. On day 30, animals were injected s.c. with 200 LD_100_ (400 CFU for mice, 6 × 10^3^ CFU for guinea pigs) of the wild-type *Y. pestis* strain 231. All animals were observed over a 30-day period.

### 2.8. ELISA

ELISA was used to assay serum IgG antibodies against recombinant LcrV [30] and F1 [31] proteins and *Y. pestis* 231 whole-cell lysate (YPL) [32]. Polystyrene 96-well flat-bottom microtiter plates (Dynatech Laboratories Inc., Chantilly, VA, USA) were coated with 1 μg/well recombinant protein or 0.5 μg/well YPL in 0.3 N sodium carbonate buffer [pH 9.6]. Mouse and guinea pig sera samples were serially diluted from 1:250 to 1:512,000. The endpoint dilution titer was calculated as the serum dilution resulting in an absorbance reading of 0.2 units above background. Goat anti-mouse IgG-HRP (1:5000, Sigma, Saint-Louis, MO, USA) and goat anti-guinea pig IgG-HRP (1:5000, Sigma, Saint-Louis, MO, USA) were used as the detection antibodies. The reactions were developed with TMB (3,3′,5,5′-Tetramethylbenzidine) and stopped with 2 M H_2_SO_4_. The absorbance at 450 nm was measured. Background values were obtained from serum samples collected from the animals injected with the PBS alone.

### 2.9. In Vivo Cytokine Analysis

Outbred mice (*n* = 3) and guinea pigs (*n* = 3) were inoculated s.c. with 10^4^ CFU of *Y. pestis* 231(P_BAD_-*crp*)/pPst¯. A group of uninfected mice served as the negative control (0 day). Samples were taken from mice inoculated s.c. on days 2, 4, and 6 post-inoculation. The sera were filtered through 0.22-μm syringe filters (PTFE-0.22-13, JINTENG, Tianjin, China) and cultured on BHI plates to confirm their sterility. The levels of cytokines were measured using an ELISA with Guinea pig Interferon-γ ELISA Kit (#EGP0017), Guinea pig TNF-α ELISA Kit (#EGP0049), Mouse IFN-γ ELISA Kit (#EM0093), and Mouse TNF-α ELISA Kit (#EM0183), all according to the manufacturer’s instructions (Fine Biotech Co., Ltd., Wuhan, China).

### 2.10. Statistics

The LD_50_ and 95% confidence intervals of the isogenic *Y. pestis* strains for mice and guinea pigs were calculated using the Kärber method [22]. *t*-test of unpaired samples, ANOVA and Log-rank (Mantel–Cox) test were used. A *p*-value below 0.05 was considered significant.

## 3. Results

### 3.1. Construction of the Deletion-Insertion Mutation to Achieve Regulated Delayed Attenuation

The Δ*crp* mutant of *Y. pestis* strain 231 was generated by replacing the *crp* with the chloramphenicol resistance gene *cat* and curing the resistance cassette to produce a FRT scar (Figure 1A). The P_BAD_ *crp* mutant of *Y. pestis* strain 231 was constructed by replacing 74 nucleotides containing the *crp* promoter region from −110 to −37 bp with the transcription terminator t0 from bacteriophage λ (Lt0TT) and the *araC* P_BAD_ promoter from *E. coli* strain DH5α (Figure 1B). The deletion of the *crp* from wild type 231 and the promoter replacement with P_BAD_-*crp* were confirmed by PCR analysis using specific primers (Table 2) as well as by DNA sequencing of the flanking regions of the *crp* on the chromosome. The above-mentioned genetic manipulations resulted in creation of the isogenic Δ*crp* and P_BAD_-*crp* mutants of *Y. pestis* strain 231.

The growth of the P_BAD_-*crp* mutant strain in the presence of arabinose leads to transcription of the *crp*, but gene expression stopped in the absence of arabinose. The presence of the Crp protein in *Y. pestis* mutant strains was tested by Western blot using mouse antibodies raised against the His-tagged recombinant protein. Crp was not detected in the either 231Δ*crp* or 231 P_BAD_-*crp* mutants grown in the absence of arabinose (Figure 2). After arabinose addition, the P_BAD_-*crp* strain synthesized approximately the same amount of Crp as the parent wild-type *Y. pestis* strain 231.

The Δ*crp* and the P_BAD_-*crp* strains without arabinose grew more slowly than the parent *Y. pestis* strain 231 in BHI medium.When 0.2% arabinose was added to the nutrient medium, P_BAD_-*crp* grew at the same rate as the wild type strain.

Crp increased expression of *malT*, the transcriptional activator of maltose metabolism [33]. The test of maltose fermentation can readily display the Crp activity. We analyzed the growth of wild-type 231, 231Δ*crp*, P_BAD_-*crp* on MacConkey maltose agar without and with 0.2% arabinose (Figure 3). The parent wild-type strain of *Y. pestis* was able to ferment maltose to acid, which was accompanied by a decrease in pH, and as a result, the colonies changed color on MacConkey medium containing a red indicator with a neutral pH. The *Y. pestis* 231Δ*crp* mutant did not grow with maltose as the sole carbon source. The *Y. pestis* 231P_BAD_-*crp* strain could ferment maltose only when grown in the presence of arabinose, but not in the absence of it.

### 3.2. Attenuation of Mutant Strains in s.c. Challenged Mice and Guinea Pigs

To investigate the contribution of Crp to *Y. pestis* 231 strain virulence, we infected outbred mice and guinea pigs s.c. with *Y. pestis* 231, 231Δ*crp*, or 231P_BAD_-*crp*. *Y. pestis* 231P_BAD_-*crp* strain was grown with arabinose prior to inoculation. The strain 231Δ*crp* was nonlethal at doses of ≤10^7^ CFU in mice. The LD_50_ of 231Δ*crp* strain in guinea pigs increased at least by 10^7^-fold (LD_50_ = 2.1 × 10^8^ CFU) compared with that of the wild-type strain (LD_50_ = 3 CFU) (Table 3). The LD_50_s of the 231P_BAD_-*crp* mutant in mice and guinea pigs were approximately 10^4^-fold and 10^7^-fold higher than those of *Y. pestis* 231, respectively.

The degree of attenuation of the 231(P_BAD_-*crp*) mutant strain was increased by eliminating the pPst plasmid. We tested the ability of the plasmid-cured strain to infect outbred mice and guinea pigs. The 231P_BAD_-*crp*(pPst¯) strain was nonlethal at doses of approximately ≤10^7^ CFU in mice and ≤10^9^ CFU in guinea pigs (Figure 4).

### 3.3. Humoral Immune Responses

Sera were collected from vaccinated mice and guinea pigs 28 days after immunization. The levels of serum IgG titers against two main immunodominant *Y. pestis* protective antigens (F1 and LcrV) and *Y. pestis* whole-cell lysate (YPL) were determined via ELISA (Figure 5). Anti-F1 (*p* < 0.0001), anti-LcrV (*p* < 0.0001), and anti-YPL (*p* < 0.0001) IgG titers in mice and guinea pigs increased in a dose-dependent manner when the levels of IgG induced by different immunized doses were compared. The differences in the levels of IgG to *Y. pestis* antigens in the sera of mice and guinea pigs immunized with 10^2^ CFU were insignificant (*p* > 0.05).

### 3.4. Cell-Mediated Immune Response

Then we measured the levels of the cytokines TNF-α and IFN-γ in sera of mice and guinea pigs at various times after immunization with *Y. pestis* strain 231P_BAD_-*crp*(pPst¯) (Figure 6). In s.c.-immunized mice and guinea pigs, induction of pro-inflammatory cytokines IFN-γ and TNF-α followed similar kinetics. The levels of cytokines increased significantly by 48 h. Then the level of IFN-γ in the serum of mice and guinea pigs and TNF-α in guinea pigs decreased. In the serum of mice, the level of TNF-α remained almost at the same level from the second to the sixth day after the immunization with *Y. pestis* strain 231P_BAD_-*crp*(pPst¯).

### 3.5. Ability of s.c. Administered Strain with araC P_BAD_-Regulated crp to Induce Protective Immunity to s.c. Challenge with Wild-Type Y. pestis Strain 231

Groups of mice and guinea pigs were subcutaneously immunized with 10-fold or 100-fold (respectively) dilution doses of *Y. pestis* strain 231P_BAD_-*crp*(pPst¯) and challenged subcutaneously 30 days later with 200 LD_100_ (400 CFU for mice, 6 × 10^3^ CFU for guinea pigs) of the wild-type strain 231.

Mice immunized with 10 CFU of the *Y. pestis* 231P_BAD_-*crp*(pPst¯) mutant succumbed to plague after being challenged with 200 LD_100_ of the WT *Y. pestis* strain 231, but the post-infection lifespan of animals in this group was significantly increased compared with that of the naïve mice. 66.6% of the mice immunized subcutaneously with 10^2^ or 10^3^ CFU survived after challenge with 200 LD_100_ of the WT *Y. pestis*. 10^2^ CFU, the lowest immunizing dose of the strain 231P_BAD_-*crp*(pPst¯), did not protect guinea pigs against infection with 200 LD_100_ of the wild type *Y. pestis* strain 231. The immunizing doses of ≥10^4^ CFU were sufficient to ensure 100% survival of both animal species. (Figure 7).

## 4. Discussion

Developing live bacterial vaccines presents a significant challenge: ensuring safety without compromising the ability to protect vaccinated individuals [34]. However, traditional attenuation methods often result in bacteria becoming overly susceptible to the host’s innate immunity, which prevents the development of a strong immune response. To overcome these limitations, researchers have engineered bacterial strains that initially exhibit the behavior of virulent wild-type bacteria, allowing them to effectively colonize the host’s tissues. Subsequently, these strains undergo a precisely controlled and delayed attenuation process within the host. This strategy prevents the disease while still triggering a powerful immune response. This regulated delayed attenuation can be accomplished through various genetic modifications [14].

One approach involves substituting promoters of essential for virulence genes (*fur, crp, phoPQ*, *rpoS*) with an arabinose-dependent *araC* P_BAD_ cassette [13]. This ensures that expression of these genes relies on the presence of available arabinose during growth. Upon colonization of host tissues, arabinose deprivation halts the synthesis of these genes’ products, leading to gradual attenuation and preventing disease manifestations.

Initially, the *araC* P_BAD_ mutant was generated by the developers of the delayed attenuation method. Its LD_50_ for mice was 10^5^ CFU [16], which indicated significant residual virulence and was not good for a vaccine strain. To further reduce virulence, the *lpxL* gene from *E. coli* was introduced into the genome of the *araC* P_BAD_ mutant. This gene is coding a late acyltransferase responsible for the addition of the sixth fatty acid residue to the lipid A. The resulting hexaacyl variant of lipid A is successfully recognized by the host immune system, which prevents the development of a lethal infection [35]. The double mutant further reduced its virulence. For mice its LD_50_ was >10^7^ CFU [36]. Deletion of the introduced gene is not so unlikely. From a biosafety point of view, it is undesirable to use such strains as vaccines, since it is impossible to exclude reverse or compensatory mutations fraught with reversion of virulence. However, a drawback of this vaccine strain variant is the potential loss of the *lpxL* gene integrated into the *Y. pestis* genome and restoration of virulence.

To achieve the same goals—further reducing virulence—we used an alternative approach: elimination of the pPst plasmid from the genome, believing that the probability of de novo emergence of the plasminogen activator gene is incomparably lower than the probability of loss of the *lpxL* gene. This plasmid encodes the well-known pathogenicity factor, plasminogen activator Pla. In the majority of studies, knockout of this gene resulted in an increase in LD_50_ values up to 10^6^ or more CFU [37], but in some experiments LD_50_ values did not increase, while the survival time of mice after infection with a Pla-negative strains slightly increased [38].

Relative evaluation of the *Y. pestis* Δ*crp* knockout mutant with its isogenic *araC* P_BAD_ *crp* variant with arabinose-dependent regulated expression of the *crp* gene showed that, compared with the parental KIM5+ strain (bv. Medievalis), both strains were significantly attenuated upon s.c. infection of Swiss Webster mice [16]. LD_50_ of the Δ*crp* (3 × 10^7^ CFU) and *araC* P_BAD_ *crp* (4.3 × 10^5^ CFU) mutants were approximately 10^6^ and 10^4^ times higher than those of the parent *Y. pestis* strain KIM5+ (<10 CFU), respectively. In our experiments conducted on derivatives of the *Y. pestis* strain 231 from another phylogenetic group (bv. Antiqua), we obtained similar results. Compared with the parent strain (LD_50_ = 3 CFU for both animals), the LD_50_ of the Δ*crp* knockout mutant increased for outbred mice and guinea pigs by approximately six and seven orders of magnitude, while the virulence of the delayed-attenuation 231P_BAD_-*crp* variant decreased by approximately 1.9 × 10^4^ and 1.5 × 10^7^ times.

However, according to the Russian national guidelines [39], *Y. pestis* candidate vaccine strains should not cause death in mice infected subcutaneously with 10^7^ CFU and guinea pigs infected subcutaneously with 2 × 10^9^ CFU and even 1.5 × 10^10^ CFU. The Δ*crp* mutant and the less attenuated isogenic P_BAD_-*crp* mutant did not meet these criteria.

At the same time, it is known that means for achieving regulated delayed attenuation can be combined with other mutations, which together may yield safe efficacious recombinant attenuated bacterial vaccines [36]. The combination of different mutagenesis methods allows uniting diverse strategies for the rational approach to design of attenuated strains of pathogenic bacteria, candidates for vaccine strains [40]. In our study, we took advantage of the fact that in some cases, the loss of the ability to produce plasminogen activator is accompanied by significant attenuation, and the plasmid encoding Pla does not contain genes for protective antigens [38]. After low-temperature cultivation, we were able to select a pPst¯ clone of P_BAD_-*crp* mutant that reduced its virulence compared to the wild-type strain by at least 7 orders of magnitude in mice and at least 9 orders of magnitude in guinea pigs.

Is the path we have chosen to develop candidate vaccine strains not too complex? Perhaps we could have stopped at one of the intermediate stages? Or, on the contrary, should we have continued editing the genome to remove or modify genes encoding proteins that are undesirable from a vaccinological perspective? Do our mutants combine safety with effective protection for immunized individuals belonging to different species or phylogenetic groups? Below we will try to answer some of these questions.

It is known that different animals react differently to *Y. pestis* antigens [41]. A mixed vaccine including the strains with special efficacy in rats and in guinea pigs protects both animal species better than the monocomponent ones. A multivalent vaccine, containing strains proven effective against both rats and guinea pigs, offers superior protection for these two animal species compared to vaccines targeting only a single strain. The efficacy of plague vaccines has been demonstrably enhanced through various combinatorial approaches. Studies have shown that administering live attenuated vaccines (#46-S, M #74, or MP-40) in conjunction with the existing EV vaccine significantly amplifies their protective effect [42].

The generation of antibodies specific to *Y. pestis* is crucial in fighting off infection [39,43,44,45,46,47]. However, a strong and effective immune response against this pathogen also requires the activation of cellular immune mechanisms [48,49,50,51]. Immunization of mice and guinea pigs with 231P_BAD_-*crp*(pPst¯) elicits anti-F1, anti-LcrV, and anti-YPL serum IgG in a dose-dependent manner. The induction of cellular immunity against *Y. pestis* by 231P_BAD_-*crp*(pPst¯) is confirmed by the observed increase in TNF-α and IFN-γ production levels in mice and guinea pigs immunized subcutaneously. Thus, *Y. pestis* 231P_BAD_-*crp*(pPst¯) seems to stimulate both humoral and cellular immunity. Mice and guinea pigs immunized subcutaneously with a single dose (10^4^ CFU) of the 231P_BAD_-*crp*(pPst¯) mutant were completely protected against a subcutaneous challenge with 200 LD_100_ of the wild type *Y. pestis* strain.

The strain 231P_BAD_-*crp*(pPst¯) is of interest as a vaccine candidate. It is possible that its predecessor with the pPst plasmid has a greater ability to protect animals due to greater residual virulence, but its ability to cause death in some infected individuals at doses of no more than 10^7^ CFU excludes it from potential vaccine candidates.

## 5. Conclusions

Our results confirm that arabinose-dependent regulated expression of *crp* in combination with elimination of the pPst plasmid is an effective strategy for attenuating *Y. pestis* while maintaining high immunogenicity, providing protection in murine and guinea pig models against bubonic plague.

## Figures and Tables

**Figure 1 biomolecules-16-00040-f001:**
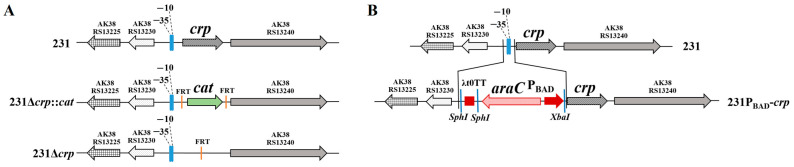
Schematic diagram depicting the chromosomal structure of the *crp* deletion mutation (**A**) and deletion-insertion mutations resulting in the arabinose-regulated *crp* gene (**B**).

**Figure 2 biomolecules-16-00040-f002:**
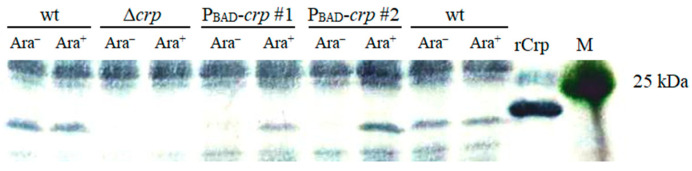
Crp synthesis in *Y. pestis crp* mutants. Strains were grown in BHI without (Ara^–^) and with 0.2% arabinose (Ara^+^) at 37 °C overnight, and Crp synthesis was detected by Western blotting using anti-Crp sera. M, protein marker; rCrp—recombinant Crp; wt, wild type. Original WB are in Appendix A.

**Figure 3 biomolecules-16-00040-f003:**
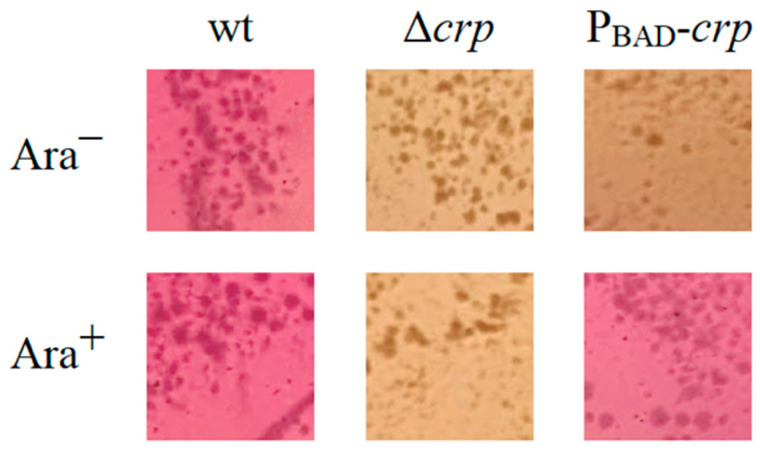
Phenotypes of *Y. pestis* strains with *crp* deletion-insertion mutations streaked on MacConkey maltose agar without (Ara^−^) and with 0.2% arabinose (Ara^+^).

**Figure 4 biomolecules-16-00040-f004:**
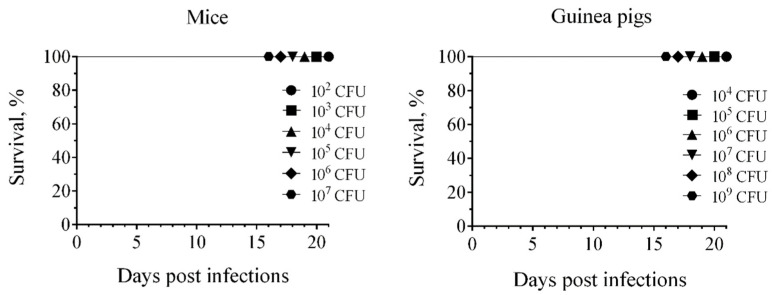
Virulence of *Y. pestis* strain 231P_BAD_-*crp*(pPst¯) for subcutaneously infected outbred mice (*n* = 6) and guinea pigs (*n* = 6).

**Figure 5 biomolecules-16-00040-f005:**
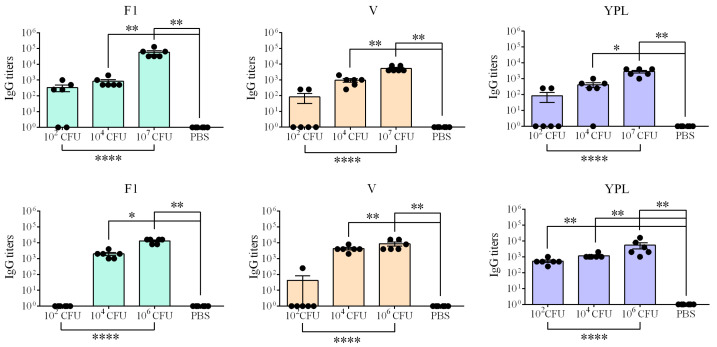
Antibody responses in sera of mice and guinea pigs immunized s.c. with 231P_BAD_-*crp*(pPst¯) at day 28 post immunization. *Y. pestis* whole-cell lysate (YPL) and recombinant F1 and LcrV were used as the coating antigens. *t*-test of unpaired samples and ANOVA were used. Antibody titers are shown as individual values with a mean (Mean) with Standard Error of Measurement (SEM). *—*p* < 0.05; **—*p* < 0.005; ****—*p* < 0.0001.

**Figure 6 biomolecules-16-00040-f006:**
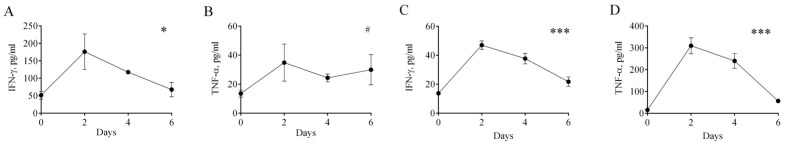
Cytokine levels in the sera of mice (**A**,**B**) and guinea pigs (**C**,**D**) immunized s.c. with 10^4^ CFU of *Y. pestis* strain 231P_BAD_-*crp*(pPst¯). Serum samples were collected from 3 mice and 3 guinea pigs on days 2, 4, and 6 p.i. Values were calculated as picograms of cytokine per ml of blood. All experiments were performed twice with similar results. ANOVA was used. Cytokines levels were shown as the mean (Mean) with Standard Error of Measurement (SEM). #—*p* > 0.05; *—*p* < 0.05; ***—*p* < 0.005.

**Figure 7 biomolecules-16-00040-f007:**
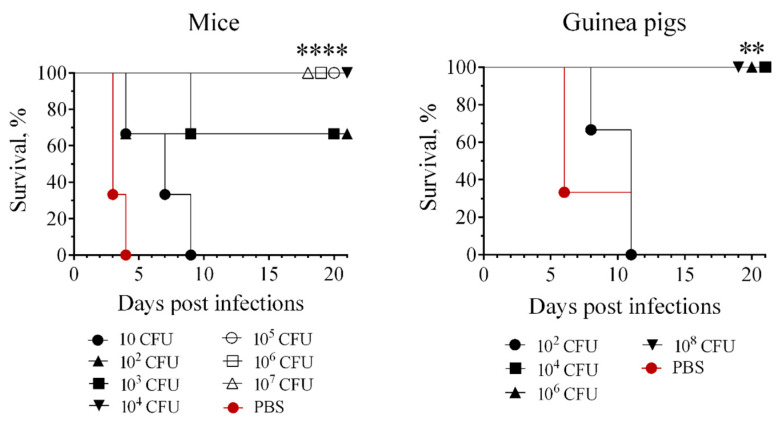
Survival of outbred mice (*n* = 6) and guinea pigs (*n* = 6) subcutaneously challenged with 200 LD_100_ of the WT *Y. pestis* strain 231 30 days after subcutaneous immunization with *Y. pestis* P_BAD_-*crp*(pPst¯) grown in the presence of arabinose. Log-rank (Mantel–Cox) test was used. **—*p* < 0.05, ****—*p* < 0.0001.

**Table 1 biomolecules-16-00040-t001:** Bacterial strains and plasmids used in this study.

Strain, Plasmid	Relevant Attributes	Source
*Y. pestis*		
231	0.ANT3 phylogroup, wild-type strain, universally virulent (LD_50_ for mice ≤ 10 CFU, for guinea pigs ≤ 10 CFU); Pgm^+^, pMT1^+^, pPst^+^, pCD^+^, parental strain	SCPM-O ** [19]
EV	1.ORI3 phylogroup, vaccine strain, Δ*pgm* *, pMT1^+^, pPst^+^, pCD^+^	SCPM-O
EVΔ*crp*::*cat*	Δ*crp* derivative of EV, Cm^R^	
231Δ*crp*	Δ*crp* derivative of 231, Cm^S^	This study
231P_BAD_-*crp*	ΔP*_crp_*::*araC* P_BAD_ *crp* derivative of *Y. pestis* 231	This study
231P_BAD_-*crp*(pPst¯)	ΔP*_crp_*::*araC* P_BAD_ *crp* pPst¯ derivative of *Y. pestis* 231P_BAD_-*crp*	This study
*E. coli*		
DH5α	F^−^, *gyrA96*(Nal^r^), *recA1*, *relA1*, *endA1*, *thi-1*, *hsdR17*(r_k_*^–^*, m_k_^+^), *glnV44*, *deoR*, Δ(*lacZYA*-*argF*)*U169*, [φ80dΔ(*lacZ*)*M15*], *supE44*	SCPM-O
S17-1 λ*pir*	*thi pro hsdR*^−^*hsd*M^+^ *recA* RP4 2-Tc::Mu-Km::Tn*7*(Tp^R^Sm^R^Pm^S^)	SCPM-O
BL21(DE3)	*F^–^ompT hsdS_B_ (r_B_^–^ m_B_^–^) gal dcm (DE3)*	SCPM-O
Plasmid		
pKD46	bla *araC* P_BAD_gam bet exo pSC101 oriTS	[20]
pKD3	*bla* FRT *cat* FRT PS1 PS2 *ori*R6K	[20]
pCP20	*bla cat c*I857 λP_R_*flp* pSC101 *ori*TS	[21]
pET-24b (+)	kan pBR322 ori P_T7_	Novagene (Madison, WI, USA; now Millipore Sigma)
pET24-*crp*	kan pBR322 ori P_T7_ *crp*	This study
pUC57	*bla* pUC *ori*	Thermo Fisher Scientific (Vilnius, Lithunia)
pUC57-URcrp-*araC* Pbad-crp	bla pUC ori *araC* P_BAD_ *crp*	This study
pUC57-URcrp-Lt0TT-*araC* P_BAD_-*crp*	bla pUC ori bacteriophage Lambda t0 transcriptional terminator P_BAD_ *crp*	This study
pCVD442	*ori* R6K *mob* RP4 *bla sacB*	[22]
pCVD442-Δ*crp*::*cat*	bla R6K ori RP4 mob sacB Δ*crp*::*cat*	This study
pCVD442-URcrp-Lt0TT- *araC* P_BAD_-*crp*	bla R6K ori RP4 mob sacB bacteriophage Lambda t0 transcriptional terminator P_BAD_ *crp*	This study

* The pigmentation locus (*pgm*) is a large, unstable fragment of the *Y. pestis* chromosome containing an iron uptake region. Loss of this locus results in a dramatic reduction in virulence [23]. ** The State Collection of Pathogenic Microbes and Cell Cultures on the base of the State Research Center for Applied Microbiology and Biotechnology (“SCPM-Obolensk”).

**Table 2 biomolecules-16-00040-t002:** Primers used in this study.

*crp* Primers for Mutant Construction and Screening	
Crp1F	ATGGTTCTCGGTAAGCCACAAACAGACCCGACTCTCGAATGGTTCCTGTCTCATTATGGGAATTAGCCATGGTCC
Crp1R	TTAACGGGTGCCGTAAACGACGATCGTTTTACCGTGTGCGGAGATCAAGTTTTGAGTGTAGGCTGGAGCTGCTTC
Crp2F	TAACAACAAAGATACAGCCC
Crp2R	AGTAACAAAATTGTGCCACC
Crp-KF	GACTTCGCGTACCTCAAAGC
Crp-KR	TACATAACCGGAACCACAAC
Primers for pCVD442-URcrp-Lt0TT-P_BAD_-crp Construction	
Pbad-SphI	GCGGCATGCATAATGTGCCTGTCAAATGG
Pbad-XbaI	GCGTCTAGAGAGAAACAGTAGAGAGTTGC
Lt0-SphIF	AGCGCATGCTGACTCCTGTTGATAGATCC
Lt0-SphIR	TTTGCATGCGACAAGTTGCTGCGATTCTC
Crp-Hind	CCTAAGCTTCCCGGGTCGGCTGATAGATCAACTGC
Crp-SphI	GAAGCATGCGCCGAAAGGTATAGCCAAGG
Crp-XbaI	GCGTCTAGAAAGTTAGGCAGCGATAACAAC
Crp-SalI	CTTGTCGACTTAACGGGTGCCGTAAAC
Primers for pET24-crp Construction	
Crp-NdeI	TAGTATCATATGGTTCTCGGTAAGCCACA
Crp-XhoI	TACTCGAGACGGGTGCCGTAAACGACGAT
Screening for pCD1	
yscFPlus	ACACCATATGAGTAACTTCTCTGGATTTACG
yscFMinus	ATTCTCGAGTGGGAACTTCTGTAGGATG
Screening for pMT	
caf1Plus	AGTTCCGTTATCGCCATTGC
caf1Minus	GGTTAGATACGGTTACGGTTAC
Screening for pPst	
PstF	CAATCATATGTCAGATACAATGGTAGTG
PstR	CTCCTCGAGTTTTAACAATCCACTATC

**Table 3 biomolecules-16-00040-t003:** Virulence of *Y. pestis* strains in subcutaneously infected outbred mice and guinea pigs.

*Y. pestis* Strains	LD_50,_ CFU	
	Mice	Guinea Pigs
231	3	3
231Δ*crp*	>10^7^	2.1 × 10^8^
231P_BAD_-*crp*	5.6 × 10^4^	4.6 × 10^7^
231P_BAD_-*crp*(pPst¯)	>10^7^	>10^9^

## Data Availability

The original contributions presented in this study are included in the article/Appendix A. Further inquiries can be directed to the corresponding author.

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
