# Peer review of "Virulence Reduction in *Yersinia pestis* by Combining Delayed Attenuation with Plasmid Curing"

_biomolecules, 2025, doi:10.3390/biom16010040_

Round 1

Reviewer 1 Report

Comments and Suggestions for Authors

The study presents a rationally designed live attenuated Y. pestis vaccine based on crp regulation and pPst curing in strain 231. The authors generate Δcrp and PBAD‑regulated mutants, confirm functional Crp control, and demonstrate strong attenuation (LD50 increases of 4–9 logs) and full protection at ≥10⁴ CFU in mice and guinea pigs. The work meets Russian safety criteria and aligns conceptually with WHO and FDA Animal Rule priorities.

Strengths:

  • Clear rationale addressing the need for safe, effective plague vaccines.

  • Sound methodology for genetic construction, virulence testing, and immunogenicity assays.

  • Convincing evidence of attenuation, protection, and combined humoral/cellular immune responses.

  • Dual-species evaluation enhances regulatory and translational relevance.

Main weaknesses and recommendations:

  • Clarify novelty versus prior crp-based KIM5⁺ work and clearly position this candidate relative to existing subunit and live vaccines (e.g., rF1V, EV).

  • Provide stability data for the PBAD‑regulated locus and pPst loss, and discuss reversion risks and safety considerations (shedding, environmental release).

  • Add or discuss quantitative correlations between immune responses and survival, plus any bacterial burden or histopathology data supporting attenuation.

  • Strengthen reporting transparency (randomization, blinding, LD100 definition) and include key numerical values in Results.

  • Streamline the Discussion to emphasize 3–4 main messages.

Overall assessment:
A solid, well‑executed study contributing incremental but meaningful progress toward rationally attenuated plague vaccines. Recommended for publication after moderate revision addressing novelty, stability/safety, and clearer contextualization.

Author Response

Thank you for your high praise of our work and constructive comments. Please find the detailed responses below.

 Comment 1: Clarify novelty versus prior crp-based KIM5⁺ work and clearly position this candidate relative to existing subunit and live vaccines (e.g., rF1V, EV).

Response 1: Interrelation of our work with prior crp-based KIM5⁺ work is now presented in the discussion section.

Initially, the araC PBAD mutant was generated by the developers of the delayed attenuation method. Its LD50 for mice was 105 CFU [Sun, W.; Roland, K.L.; Kuang, X.; Branger, C.G.; Curtiss, R. yersinia pestis with regulated delayed attenuation as a vaccine candidate to induce protective immunity against plague. Infect Immun 2010, 78, 1304–1313, doi:10.1128/IAI.01122-09], which indicated significant residual virulence and was not good for a vaccine strain. To further reduce virulence, the lpxL gene from E. coli was introduced into the genome of the araC PBAD mutant. This gene is coding a late acyltransferase responsible for the addition of the sixth fatty acid residue to the lipid A. The resulting hexaacyl variant of lipid A is successfully recognized by the host immune system, which prevents the development of a lethal infection [Montminy SW, Khan N, McGrath S, et al. Virulence factors of Yersinia pestis are overcome by a strong lipopolysaccharide response. Nat Immunol. 2006;7(10):1066-1073. doi:10.1038/ni1386]. The double mutant further reduced its virulence. For mice its LD50 was >107 CFU[ [Sun W, Six D, Kuang X, Roland KL, Raetz CR, Curtiss R 3rd. A live attenuated strain of Yersinia pestis KIM as a vaccine against plague. Vaccine. 2011;29(16):2986-2998. doi:10.1016/j.vaccine.2011.01.099]. Deletion of the introduced gene is not so unlikely. From a biosafety point of view, it is undesirable to use such strains as vaccine ones, since it is impossible to exclude reverse or compensatory mutations fraught with reversion of virulence. However, a drawback of this vaccine strain variant is the potential loss of the lpxL gene integrated into the Y. pestis genome and restoration of virulence.

To achieve the same goals—further reducing virulence—we used an alternative approach: elimination of the pPst plasmid from the genome, believing that the probability of de novo emergence of the plasminogen activator gene is incomparably lower than the probability of loss of the lpxL gene. This plasmid encodes the well-known pathogenicity factor, plasminogen activator Pla. In the majority of studies, knockout of this gene resulted in an increase in LD50 values up to 106 or more CFU [Sodeinde, O.A.; Subrahmanyam, Y.V.; Stark, K.; Quan, T.; Bao, Y.; Goguen, J.D. A surface protease and the invasive character of plague. Science 1992, 258, 1004–1007. doi: 10.1126/science.143979], but in some experiments LD50 values did not increase, while the survival time of mice after infection with a Pla-negative strains slightly increased [Sebbane F, Uversky VN, Anisimov AP. Yersinia pestis Plasminogen Activator. Biomolecules. 2020;10(11):1554. Published 2020 Nov 14. doi:10.3390/biom10111554].

As for comparisons with other plague vaccines, there is currently one licensed live plague vaccine based on several lines of the Y. pestis EV76 strain, and approximately twenty candidate vaccines are undergoing preclinical and clinical trials. We plan to conduct comparative studies of our strain with other candidate vaccines, but this will be the subject of a separate study.

Comment 2: Provide stability data for the PBAD‑regulated locus and pPst loss, and discuss reversion risks and safety considerations (shedding, environmental release).

Response 2: The araC PBAD system is regulated by the positive regulator AraC, which is the product of the araC gene. The araC mutants that are unable to produce normal AraC protein cannot provide gene expression from the PBAD promoter [Irr J, Englesberg E. Nonsense mutants in the regulator gene araC of the L-arabinose system of Escherichia coli B-r. Genetics. 1970 May;65(1):27-39. doi: 10.1093/genetics/65.1.27. PMID: 4920896; PMCID: PMC1212430.]. Any mutation that can occur in this system can only lead to the absence of expression of the gene it controls, and not to its constitutive expression. Thus, reversion to a virulent phenotype by resuming expression of the target gene in the absence of arabinose is impossible.

Comment 3: Add or discuss quantitative correlations between immune responses and survival, plus any bacterial burden or histopathology data supporting attenuation.

Response 3: In our case, levels of specific antibodies to the capsular antigen were weakly correlated with the survival of mice (r=0,2356) and moderately for guinea pigs (r=0,4252). Mice survival moderately correlated with anti-V antibody levels (r=0,4175) and evidently for guinea pigs (r=0,5028).

Regarding bacterial burden or histopathology data, such studies have not been conducted. This article presents data from preliminary experiments. In the future, we plan to conduct more comprehensive studies of the constructed strains.

Comment 4: Strengthen reporting transparency (randomization, blinding, LD100 definition) and include key numerical values in Results.

Response 4: Regarding this reviewer's comment, the data he was interested in was provided to the editors in the form "The ARRIVE guidelines 2.0: author checklist" and, presumably, it can be provided to reviewers upon request.

Comment 5: Streamline the Discussion to emphasize 3–4 main messages.

Response 5: This reviewer's comment about reducing the number of main messages to 3-4 contradicts the need to respond to the remaining reviewer comments. Replies to these comments lead to the expansion of the Discussion section. We hope our reviewers will forgive us for this.

Reviewer 2 Report

Comments and Suggestions for Authors

Manuscript: "Virulence reduction in Yersinia pestis by combining delayed attenuation with plasmid curing"

The present study is a well-designed and methodologically sound investigation into the application of a novel combinatorial approach for attenuating Yersinia pestis to develop vaccines. The authors have successfully combined two attenuation strategies, regulated delayed attenuation of the crp gene with curing of the pPst plasmid, to generate a strain that is both safe and immunogenic in two animal models, mice and guinea pigs. The work is scientifically sound, addresses an important need for public health, and provides compelling data that support the potential of the engineered strain as a plague vaccine candidate. The manuscript is well-written in general, logically arranged, and supported by appropriate controls and statistical analyses. It is suitable for publication with minor revisions.

Minor suggestions for improvement:

The introduction might provide a clearer hypothesis, such as how the combination of delayed crp attenuation with pPst curing will yield a safer yet still immunogenic vaccine candidate compared to single modifications.

Consider adding a brief rationale for why pPst was chosen for curing (e.g., its role in virulence but not in protective antigenicity).

Statistical reporting:

Error bars and statistical test details are mentioned in legends, yet the measure specifics in some figures (e.g., Figure 5, 6) are not defined, such as SD vs. SEM. Please clarify in legends.

For survival curves (Figure 7), include the number of animals per group in the legend or figure caption.

Discussion:

The discussion contextualises the findings thoughtfully, but could be strengthened by:

Comparing the immunogenicity of the double mutant directly with the single mutants (e.g., antibody titers, cytokine levels).

Speculating on the reasons why pPst curing further attenuates without compromising protection-reducing inflammation or altering antigen presentation?

Addressing limitations, such as the use of only a subcutaneous challenge; future work may need to be conducted in pneumonic plague models.

Technical clarifications:

In Table 1, define what "Pgm⁺" and "Pgm⁻" mean for readers who are unfamiliar with the virulence markers of Y. pestis.

Briefly explain why low-temperature passage was used to cure pPst, including reference to the mechanism (e.g. plasmid instability at low temperatures).

Figure 2: Label lanes more clearly, such as "231PBAD-crp +Ara" and "–Ara", to immediately convey the information.

Language and flow:

The manuscript is well-written, with long and sometimes convoluted sentences-in particular, in the Abstract. Consider breaking these into shorter sentences for clarity.

Gene notation should be used consistently throughout, such as crp versus crp gene.

References and citations:

All references seem to be appropriate and up to date. Check that all citations in the text match correctly to the reference list. Reference 6 is a URL; please provide a standard citation format, if available.

Author Response

Thank you for your high praise of our work and constructive comments. Please find the detailed responses below.

Minor suggestions for improvement:

Comment 1: The introduction might provide a clearer hypothesis, such as how the combination of delayed crp attenuation with pPst curing will yield a safer yet still immunogenic vaccine candidate compared to single modifications.

Response 1: We discuss this issue in the Discussion section.

Comment 2: Consider adding a brief rationale for why pPst was chosen for curing (e.g., its role in virulence but not in protective antigenicity).

Response 2: Plasminogen activator Pla is an outer membrane protease with adhesive and invasive activities. Az a rule its removal from bacterial cell reduced virulence improving the safety of attenuated Y. pestis strains, enabling the development of a safe live plague vaccine.

Statistical reporting:

Comment 3: Error bars and statistical test details are mentioned in legends, yet the measure specifics in some figures (e.g., Figure 5, 6) are not defined, such as SD vs. SEM. Please clarify in legends.

Response 3: Corrected.

Antibody titers are shown as individual values with a mean (Mean) with Standard Error of Measurement (SEM) (Figure 5).

Cytokines levels were shown as the mean (Mean) with Standard Error of Measurement (SEM) (Figure 6).

Comment 4: For survival curves (Figure 7), include the number of animals per group in the legend or figure caption.

Response 4: The number of animals/groups in Figure 7 has been added.

Discussion:

Comment 5: The discussion contextualises the findings thoughtfully, but could be strengthened by:

  • Comparing the immunogenicity of the double mutant directly with the single mutants (e.g., antibody titers, cytokine levels).
  • Speculating on the reasons why pPst curing further attenuates without compromising protection-reducing inflammation or altering antigen presentation?

Lines 367-370:

  • Addressing limitations, such as the use of only a subcutaneous challenge; future work may need to be conducted in pneumonic plague models.

Response 5:

  • A direct comparison of the immunogenicity of the double mutant with single mutants was not performed due to the insufficient level of attenuation of the latter.
  • According to various authors, the products of the pPst plasmid do not have significant protectivity for mice, rats, guinea pigs and monkeys [Erova TE, Rosenzweig JA, Sha J, Suarez G, Sierra JC, Kirtley ML, van Lier CJ, Telepnev MV, Motin VL, Chopra AK. Evaluation of protective potential of Yersinia pestis outer membrane protein antigens as possible candidates for a new-generation recombinant plague vaccine. Clin Vaccine Immunol. 2013;20(2):227–38.].
  • We agree that future studies to determine the protective efficacy of the strain in a pneumonic plague model are needed.

Technical clarifications:

Comment 6: In Table 1, define what "Pgm⁺" and "Pgm⁻" mean for readers who are unfamiliar with the virulence markers of Y. pestis.

Response 6: Changes have been made to the table.

Comment 7: Briefly explain why low-temperature passage was used to cure pPst, including reference to the mechanism (e.g. plasmid instability at low temperatures).

Response 7: This is a tribute to tradition. We successfully used this technique at the end of the last century to obtain isogenic strains differing in antigenic composition. The method is simple, reproducible, and inexpensive. The only drawback is its time-consuming nature.

This is a tribute to tradition. We successfully used this technique at the end of the last century to obtain isogenic strains differing in antigenic composition. The method is simple, reproducible, and inexpensive. The only drawback is its time-consuming nature.

The mechanism of plasmid loss is unknown to us.

Comment 8: Figure 2: Label lanes more clearly, such as "231PBAD-crp +Ara" and "–Ara", to immediately convey the information.

Response 8: Corrected.

Language and flow:

Comment 9: The manuscript is well-written, with long and sometimes convoluted sentences-in particular, in the Abstract. Consider breaking these into shorter sentences for clarity.

Response 9: The longest sentence has been removed from the abstract.

Comment 10: Gene notation should be used consistently throughout, such as crp versus crp gene.

Response 10: We have chosen one of the options you suggested.

References and citations:

Comment 10: All references seem to be appropriate and up to date. Check that all citations in the text match correctly to the reference list. Reference 6 is a URL; please provide a standard citation format, if available.

Response 10: Corrected.

Reviewer 3 Report

Comments and Suggestions for Authors

The authors evaluated three Y. pestis strains in mice and guinea pigs. The authors demonstrated the decreased LD50 of mutants and capability of inducing protective immunity in both animals. The authors demonstrated that single 231PBAD-crp(pPst¯) injection can protect mice and guinea pigs from 200 LD100 of the wild type Y. pestis strain. Although interesting, some parts require some clarification before publication. Specific comments follow.

Major points:

  1. Line 94, section 2.2.: Please indicate sex of the animals.
  2. Line 149, section 2.5.: Please describe more details so that the readers can follow your methods. How did you induce His-Crp, how did you extract His-Crp from cells, what were the buffers used, how much protein and how many times injected, any adjuvant, etc.
  3. Line 169, section 2.7.: Please describe more details so that the readers can follow your methods. Details of buffer, volume incubation time, washing condition, etc. Also, please. indicate the source of antigens used for ELISA.
  4. Line 175, section 2.8.: Please indicate the number of animals used. Also please indicate the volume of sera and details of the syringe filters.
  5. Section 2.: Please describe the method for Western blotting.
  6. Table 3: Please explain the formula in parentheses.
  7. Figure 4: Please include graphs of bodyweight change to Figures 4 & 7.
  8. Please discuss about the durability of the vaccine.

Minor points:

  1. Please add error bars in Figures 5 and please explain error bars in Figure 6.

Author Response

Thank you for your high praise of our work and constructive comments. Please find the detailed responses below.

Major points:

Comment 1: Line 94, section 2.2.: Please indicate sex of the animals.

Response 1: Changed.

Comment 2: Line 149, section 2.5.: Please describe more details so that the readers can follow your methods. How did you induce His-Crp, how did you extract His-Crp from cells, what were the buffers used, how much protein and how many times injected, any adjuvant, etc.

Response 2: Added.

Comment 3: Line 169, section 2.7.: Please describe more details so that the readers can follow your methods. Details of buffer, volume incubation time, washing condition, etc. Also, please. indicate the source of antigens used for ELISA.

Response 3: Corrected.

Comment 4: Line 175, section 2.8.: Please indicate the number of animals used. Also please indicate the volume of sera and details of the syringe filters.

Response 4: Corrected.

Comment 5: Section 2.: Please describe the method for Western blotting.

Response 5: Added.

Comment 6: Table 3: Please explain the formula in parentheses.

Response 6: We have removed the "formulas" that are confusing you.

Comment 7: Figure 4: Please include graphs of bodyweight change to Figures 4 & 7.

Response 7: Animal weighing experiments are planned for full-scale preclinical trials.

Comment 8: Please discuss about the durability of the vaccine.

Response 8: Determining the durability of the vaccine is one of the mandatory stages of preclinical trials and will be carried out after official permission to conduct such studies will be granted.

Minor points:

Comment 9: Please add error bars in Figures 5 and please explain error bars in Figure 6.

Response 9: Corrected.

Round 2

Reviewer 3 Report

Comments and Suggestions for Authors

The authors have substantially revised their manuscript and addressed mostof the queries. Few minor points:

Line 167: Please add substrate.

Line201: Please indicate company and catalogue number of the filter.

Author Response

Dear reviewer, thank you very much for your thorough analysis and high evaluation of our manuscript. We have made the following corrections in accordance to the requests:

Comments 1: Line 167: Please add substrate.

Response 1: Added

Proteins were visualized with DAB substrate kit (Abcam, Cambridge, UK) and photographed using an Amersham ImageQuant 800 (Cytiva, USA) gel imaging system.

Comments 2: Line201: Please indicate company and catalogue number of the filter.

Response 2: Added

(PTFE-0.22-13, JINTENG, Tianjin, China)